# SPATIO-SPECTRAL SEQUENCE PROCESSING

**Nikita Kostin,**[*] **Simon Geisler,**[†] **Arthur Kosmala, Stephan Günnemann**
School of Computation, Information and Technology & Munich Data Science Institute
Technical University of Munich
{n.kostin, s.geisler, a.kosmala, s.guennemann}@tum.de

## ABSTRACT

Long-sequence data have become ubiquitous in the last decade. Transformers, being the de facto standard for processing sequences, suffer from quadratic complexity in both memory and running time, which makes them prohibitively expensive for long sequences. To augment recurrent models with efficient global information exchange, we follow the recent developments in usage of spectral information for graph neural networks (GNNs). This paper introduces Spatio-Spectral Sequence (S2Seq) models, which augment arbitrary sequence architectures with a learnable spectral branch to capture global geometric structure. Using Long-Range Arena classification benchmark, we demonstrate that our approach can yield meaningful improvements, sometimes bridging the gap to the state-of-the-art performance. We also show that truncated and even approximated spectra can provide enough information to match the performance of a full FFT calculation. Furthermore, we propose a proof-of-concept extension of S2Seq for autoregressive prediction using recurrent window updates in subquadratic time.

## 1   INTRODUCTION

The ability to process very long sequences is crucial to a wide range of modern machine learning applications, including language modeling Achiam et al. (2023), genomics (Jumper et al. (2021), Abramson et al. (2024)), or audio transcription Radford et al. (2023). Recent advances in graph learning demonstrate that combining local spatial propagation with global spectral information can significantly improve long-range performance. Spatio-Spectral Graph Neural Networks (S2GNNs) Geisler et al. (2024) show that injecting low-frequency spectral signals helps relieve purely spatial models from modeling global structure, improving both optimization and expressivity. From the geometric perspective, sequences can naturally be interpreted as chain graphs, whose Laplacian eigenvectors form a natural basis. This suggests that similar principles can be transferred into the sequence domain. Recent architectures support this intuition, with Mamba Gu & Dao (2023) and Hyena Poli et al. (2023) combining local operators with mechanisms that efficiently capture global structure.

Following S2GNNs, this work introduces Spatio-Spectral Sequence models (S2Seq), a lightweight architectural add-on that augments a chosen sequence model with a learnable spectral branch. The spectral branch captures the global structure of the sequence via spectral-domain transformations, while the baseline sequence model focuses on local and medium range dependencies. This modular design allows S2Seq to be combined with a wide range of architectures. In this sense, S2Seq is best viewed as an extension that adds a small number of parameters and therefore limited additional computation. Furthermore, in contrast to SSMs, S2Seq does not require the baseline model to be recurrent, which makes it more flexible in its modeling.

The contribution of this paper is threefold:

1. We introduce the S2Seq architecture and evaluate it using the Long-Range Arena benchmark Tay et al. (2020), demonstrating noticeable performance gains, and, in some architecture/dataset combinations, bridging the gap to state-of-the-art results.

---

[*]Now at Google.
[†]Now at Google Research.

2. We show that it is enough to compute the spectrum in a truncated or even approximate form to achieve results comparable to the full computation, and perform ablation studies on the number of frequencies required.

3. We demonstrate that S2Seq can be extended for autoregressive prediction and propose three proof-of-concept methods to achieve this.

## 2 RELATED WORK

**Sequence models.** Sequence models have progressed from RNNs (Hochreiter & Schmidhuber (1997), Cho et al. (2014)) to transformers Vaswani et al. (2017) and, more recently, to state-space models (SSMs) (Gu et al. (2021), Smith et al. (2022), Gu & Dao (2023), Dao & Gu (2024), Fu et al. (2022), Poli et al. (2023) Massaroli et al. (2023)). LSTMs Hochreiter & Schmidhuber (1997) and GRUs Cho et al. (2014), while alleviating vanishing gradients, are difficult to parallelize and struggle with long-range dependencies. Transformers address parallelization, but incur quadratic costs in $L$ due to the attention mechanism. Modern SSMs replace or augment attention with efficient long convolutions and gating, achieving subquadratic processing time, but still do not provide a generic plug-in augmentation framework.

**FFT-based architectures.** FFT-based mechanisms appear in sequence models in two common roles: as global, parameter free token mixers and as fast backends for very long 1D convolutions. These approaches offer subquadratic scaling in sequence length (typically $O(L \log L)$ in sequence length $L$, up to factors in hidden size). On the mixing side, FNet Lee-Thorp et al. (2021) replaces self-attention with a Fourier transformation applied along the token and feature axes, which provides a cheap unparameterized interaction pattern. On the long convolution side, SSMs such as S4 Gu et al. (2021) evaluate long linear time-invariant convolutions using FFTs. Similarly, Hyena Poli et al. (2023) learns long filters in the time domain using a small network and windowing, applying them as long FFT-based convolutions.

## 3 SEQUENCE CLASSIFICATION

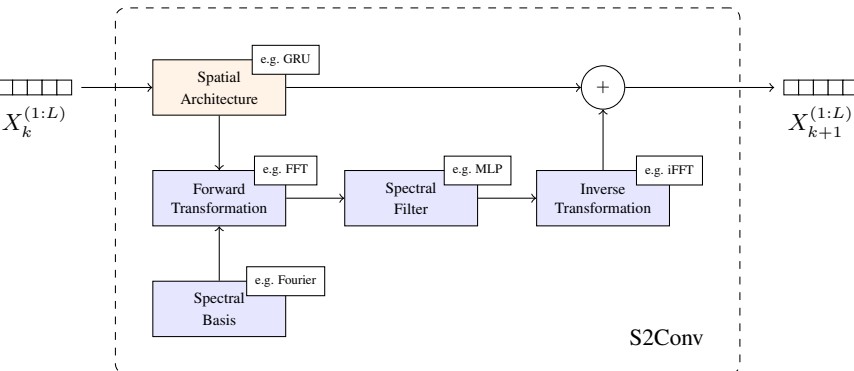

Figure 1: An overview of the S2Seq architecture containing the spatial branch and the spectral branch. $X_k^{(1:L)}$ denotes the input to the $k$-th layer of length $L$. Example implementations of each component are reported in annotation boxes. After the final layer we apply average pooling and a classification head to retrieve the final logits. Appendix A provides more details.

Our architecture adapts the core ideas of S2GNNs for sequence data. In particular, we design S2Seq to be a modular extension that can be applied on top of an arbitrary spatial architecture. The central building block of the architecture is an S2Conv layer (see Figure 1), which combines the spatial architecture with a spectral branch. In each layer, our architecture (i) processes the sequence in the time domain, (ii) maps its output into the frequency domain with a forward spectral transformation, (iii) applies a learnable filter on the spectral representation, and (iv) projects the result back via an inverse spectral transformation. The spatial and spectral outputs are then combined through a

Table 1: LRA benchmark results (↑). ✗ illustrates that the model did not exceed random guessing. Such results are excluded from the average computation. Baseline results in this table are a compilation of the results from Gu et al. (2021) and Smith et al. (2022). ↑ indicates that S2Seq provided a noticeable performance improvement over the baseline. The full table is provided in Appendix C .

| Architecture | ListOps | Text | Retrieval | Image | Path | Avg. |
|---|---|---|---|---|---|---|
| Transformer | 36.37 | 64.27 | 57.46 | 42.44 | 71.40 | 54.39 |
| BigBird | 36.05 | 64.02 | 59.29 | 40.83 | 74.87 | 55.01 |
| S4 | 59.60 | 86.62 | 90.90 | **88.65** | 94.20 | 83.99 |
| S5 | **62.15** | **89.31** | **91.40** | 88.00 | **95.33** | **85.24** |
| S2Seq[Conv1D] | 46.44 ↑ | 85.03 | 81.17 ↑ | 55.09 | ✗ | 66.93 |
| Conv1D | 33.70 | 84.59 | 74.58 | 54.29 | ✗ | 61.79 |
| S2Seq[GRU] | 60.30 ↑ | 83.40 | 84.53 ↑ | 65.33 | 73.20 | 73.35 |
| GRU | 38.70 | 82.10 | 82.89 | 65.28 | 73.85 | 68.56 |

residual connection, controlled with an optional learnable gate that adjusts the spectral contribution. Appendix A provides a more formal definition of the update rules.

We consider several spatial architectures, including one-dimensional convolutions, RNNs, and SSMs. In particular, one-dimensional convolutions can be interpreted as message passing on a chain graph, which corresponds to the graph domain setting of S2GNNs.

We further look at several orthogonal bases for representing the spectral information, including DFT Cooley & Tukey (1965), DST Ahmed et al. (2006) (DST-I), and DCT Ahmed et al. (2006) (DCT-II). As can be seen in Appendix D, the case of DST-I directly corresponds to using S2GNNs for undirected chain graphs. Despite having different boundary assumptions, empirical performance demonstrated no noticeable difference on real-world datasets when using various transformations or their combinations. Consequently, DFT is further used as the chosen spectral transformation.

Analogous to S2GNN observations, we find that only a small number of low-frequency components are sufficient to capture most of the global geometric structure. To compute partial spectra efficiently, we consider Goertzel Goertzel (1958) algorithm as well as pruned FFT variants (Markel (2003), Nagai (2003), Sorensen & Burrus (2002), Ailon & Liberty (2009), Park et al. (2020)), enabling linear or near-linear evaluation of the spectral branch. We further take a look at usage of Sparse FFT methods (Hassanieh et al. (2012), Indyk et al. (2014), Kapralov (2016), Kapralov et al. (2019)) that approximate dominant frequencies of a signal in sublinear time.

## 4 CLASSIFICATION RESULTS

We evaluate S2Seq on the Long-Range Arena (LRA) benchmark Tay et al. (2020), which comprises five long-sequence classification tasks: mathematical reasoning (ListOps), text classification, document retrieval, image classification and pathfinding. For a more complete description of the datasets please refer to Appendix B. We consider three spatial architectures (one-dimensional convolutions, GRU, and Mamba in Appendix C), and compare each S2Seq variant against its non-spectral baseline. Table 1 reports

| Dataset | # seq. | Avg. $L$ | # classes |
|---|---|---|---|
| ListOps | 100000 | 1036.96 | 10 |
| Text | 50000 | 1285.20 | 2 |
| Retrieval | 182613 | 3995.71 | 2 |
| Image | 60000 | 1024 | 10 |
| Path | 200000 | 1024 | 2 |

Table 2: LRA benchmark dataset statistics.

a truncated version of the results due to space constraints, full version of the table can be found in Appendix C. Table 2 provides statistics for LRA datasets.

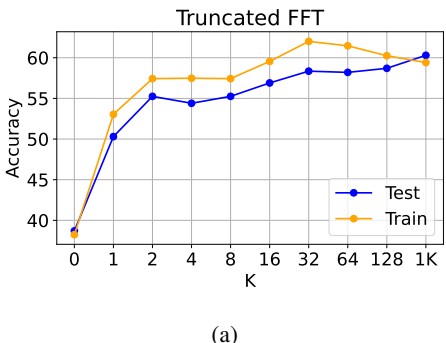 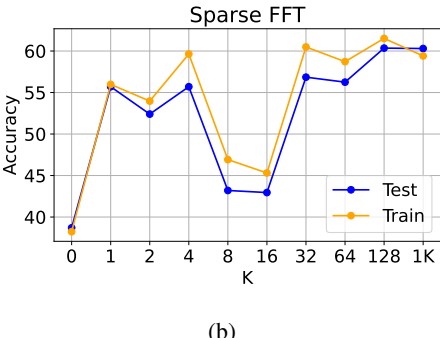

(a)                                                    (b)

Figure 2: Influence of FFT spectrum size on train and test performance on `ListOps`. (a) demonstrates the results using $K$ lowest frequency components, (b) the results using largest $K$ components per magnitude.

As can be seen in the results table, addition of the spectral information provides consistent average gains for all baseline architectures, with the strongest effect observed on `ListOps` and `Retrieval`. In particular, augmenting GRU brings its performance to near state-of-the-art results.

In Figure 2 we ablate the number of required frequencies used in the spectral branch on the `ListOps` dataset. In both truncated and sparse FFT, which in this study are emulated by selecting $K$ lowest frequency components and $K$ largest-magnitude components respectively, even a single retained frequency provides a noticeable improvement over the baseline results. Even given that $L \approx 1000$, performance reaches near-optimal levels at $K = 32$ frequencies, after which only marginal gains are achieved. For sparse FFT the convergence is less stable and exhibits a dip between $K = 8$ and $K = 16$. This behavior is consistent with magnitude-based frequency selection, which does not necessarily correlate with low-frequency components.

The shared convergence trend suggests that similar to S2GNNs, most of the gains come from capturing global structure of the sequence, while finer details are sufficiently modeled by the spatial architecture. This in particular enables efficient spectrum computation. Appendix F provides a short complexity analysis.

## 5 AUTOREGRESSIVE MODELING

In an autoregressive setting, S2Seq needs to face two complications: (i) the spectral information must respect causality ("cannot see the future"), and (ii) the current spectrum needs to be constantly updated. A naive approach would recompute the spectrum for every prefix, however doing so yields a similar or even worse complexity than the quadratic effort typically associated with transformers. This may partly explain why spectral approaches have so far seen limited adoption in autoregressive sequence modeling.

To avoid expensive spectrum recomputation we adopt a windowed autoregressive variant of S2Seq that maintains a running summary over the global geometric context and uses a partial inverse transformation for the current step. In particular, as a proof of concept, we propose the following three approaches: (i) sliding window Springer (1991), which keeps track of a fixed-length window of recent tokens, (ii) sliding window with doubling, which extends the sliding window updates with exponentially spaced reset points, and (iii) exponential window Grado et al. (2017), which replaces the hard window boundaries with an exponentially decaying kernel over past elements, effectively maintaining a soft infinite-length spectrum. Further details and update rules are provided in Appendix E.

## 6 CONCLUSION

In this work, we proposed S2Seq, a lightweight mechanism for injecting global spectral information into arbitrary sequence models. For classification, we showed that the spectral information can be

computed in sublinear time and demonstrated empirical performance gains on the LRA benchmark. We demonstrated that the frequency spectrum can be computed in truncated and approximate forms, and performed ablation studies showing that $K = 32$ frequencies is enough to demonstrate almost optimal performance on sequences with $L \approx 1000$. We further proposed a proof-of-concept extension of S2Seq approach for the autoregressive case. We hope that our research contributes to the field of spatio-spectral data processing and encourages further work on embedding spectral information into machine learning architectures.

USE OF LARGE LANGUAGE MODELS

We used LLMs to help us polish the final text of the paper and improve its readability as well as for generating boilerplate code for the figures and plots presented here. All of the ideas and contributions are our own.

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

## A  FORMAL DEFINITION OF THE CLASSIFICATION ARCHITECTURE

First, we define the spatial branch as $\text{Spatial}^{(l)}(H^{(l-1)}) = \mathcal{B}_\phi^{(l)}(H^{(l-1)})$. Here, $\mathcal{B}_\phi^{(l)}$ is an element from the family of admissible spatial architectures $\mathcal{G}$. The definition of the spectral branch depends on the forward transformation $F$ and its reverse $F^{-1}$, a point-wise embedding transformation $f_\theta$, learnable filter $\hat{g}_\vartheta$, as well as its corresponding basis eigenvalues (e.g the Fourier transform). Let $S^{(l)} = \text{Spatial}^{(l)}(H^{(l-1)})$, then

$$\text{Spectral}^{(l)}(S^{(l)}; F, \lambda) = F^{-1}\left(\hat{g}_\vartheta^{(l)}(\lambda) \odot [(F \circ f_\theta^{(l)})(S^{(l)})]\right) \tag{1}$$

Note that the input to the spectral branch is not $H^{(l-1)}$, but the output of the spatial branch. The results of both branches are then combined using a residual connection. In addition, we gate the output of the spectral branch with a learnable parameter $\gamma^{(l)}$ that helps the model adapt the intensity of spectral information for different baseline architectures and datasets:

$$H^{(l)} = \text{Spatial}^{(l)}(H^{(l-1)}) + \gamma^{(l)}\text{Spectral}^{(l)}(S^{(l)}; F, \lambda) \tag{2}$$

To produce the final logits $\overline{h}$, the model applies average pooling over the output of equation 2 and lets it pass through an MLP, i.e. $\overline{h} = (\text{MLP} \circ \text{Pool}_{\text{avg}})(H)$.

## B  EXPERIMENT SETUP

In this section we discuss the setup of the empirical evaluation and provide more detailed dataset descriptions. All experiments were executed on a single computing node with an Intel Xeon E5-2630 v4 @ 2.20GHz CPU, a single NVIDIA H100 GPU with 80GB of VRAM, and 128GB of RAM. Table 3 lists some important hyperparameters used for training.

1. **(Long) ListOps:** The sequences in this dataset are based on the original ListOps dataset Nangia & Bowman (2018) and consist of numbers, brackets, and four operators (`MAX`, `MEAN`, `MEDIAN`, `SUM_MOD`). The goal is to predict the result of the operation modulo 10.

2. **Text classification:** The task is to predict the positive/negative sentiment on long documents based on the IMDb reviews dataset Maas et al. (2011).

3. **Retrieval:** Given a pair of sequences, the objective is to determine whether they are semantically related, which requires the model to represent and compare information across both inputs. This benchmark is derived from the AAN dataset Radev et al. (2013).

4. **Image classification:** Given flattened one-dimensional images from the CIFAR-10 dataset Krizhevsky et al. (2009), the goal is to predict the class of the resulting sequence.

5. **Pathfinder:** This pathfinder challenge (Linsley et al. (2018), Kim et al. (2019)) requires the model to determine whether two marked points in a grid-based image are connected with a continuous path. Note that the `Path-X` dataset is omitted from our evaluation due to high computation cost.

Table 3: Some important hyperparameters used for the reported results. $H$ corresponds to the hidden dimension of the network. LR corresponds to learning rate. We further assume the kernel size $k = 3$ for the 1D convolution models.

| Dataset | Spatial | Spectral | Gated? | Depth | $H$ | Dropout | LR | Epochs |
|---|---|---|---|---|---|---|---|---|
| ListOps | Conv1D | DFT | ✓ | 3 | 32 | 0.0 | 1e-3 | 500 |
| | GRU | DFT | ✗ | 5 | 32 | 0.0 | 1e-3 | 400 |
| | Mamba | DFT | ✓ | 3 | 32 | 0.0 | 1e-3 | 450 |
| Text | Conv1D | DFT | ✓ | 3 | 64 | 0.1 | 1e-3 | 300 |
| | GRU | DFT | ✗ | 3 | 32 | 0.0 | 1e-3 | 200 |
| | Mamba | DFT | ✓ | 2 | 32 | 0.2 | 1e-3 | 400 |
| Retrieval | Conv1D | DFT | ✓ | 3 | 32 | 0.0 | 1e-3 | 100 |
| | GRU | DFT | ✓ | 3 | 32 | 0.0 | 1e-3 | 100 |
| | Mamba | DFT | ✓ | 2 | 32 | 0.0 | 1e-3 | 100 |
| Image | Conv1D | DFT | ✓ | 3 | 64 | 0.1 | 1e-3 | 200 |
| | GRU | DFT | ✗ | 5 | 16 | 0.0 | 1e-3 | 200 |
| | Mamba | DFT | ✓ | 3 | 32 | 0.0 | 5e-4 | 200 |
| Path | Conv1D | DFT | ✓ | 3 | 32 | 0.0 | 1e-3 | 500 |
| | GRU | DFT | ✗ | 3 | 32 | 0.0 | 1e-3 | 500 |
| | Mamba | DFT | ✓ | 3 | 32 | 0.0 | 2e-3 | 200 |

## C  ADDITIONAL RESULTS

Table 4: Full LRA benchmark results (↑). ✗ illustrates that the model did not exceed random guessing. Such results are excluded from the average computation. Baseline results in this table are a compilation of the results from Gu et al. (2021) and Smith et al. (2022). ↑ indicates that S2Seq provided a noticeable performance improvement over the baseline.

| Architecture | ListOps | Text | Retrieval | Image | Path | Avg. |
|---|---|---|---|---|---|---|
| Transformer | 36.37 | 64.27 | 57.46 | 42.44 | 71.40 | 54.39 |
| Reformer | 37.27 | 56.10 | 53.40 | 38.07 | 68.50 | 49.69 |
| BigBird | 36.05 | 64.02 | 59.29 | 40.83 | 74.87 | 55.01 |
| LinearTrans | 16.13 | 65.90 | 53.09 | 42.34 | 75.30 | 50.55 |
| Performer | 18.01 | 65.40 | 53.82 | 42.77 | 77.05 | 51.41 |
| S4 | 59.60 | 86.62 | 90.90 | 88.65 | 94.20 | 83.99 |
| S4D-LegS | 60.47 | 86.18 | 89.46 | 88.19 | 93.06 | 83.47 |
| S4-LegS | 59.60 | 86.82 | 90.90 | 88.65 | 94.20 | 84.03 |
| Liquid-S4 | **62.75** | 89.02 | 91.20 | **89.50** | 94.80 | **85.45** |
| S5 | 62.15 | **89.31** | **91.40** | 88.00 | **95.33** | 85.24 |
| S2Seq[Conv1D] | 46.44 ↑ | 85.03 | 81.17 ↑ | 55.09 | ✗ | 66.93 |
| Conv1D | 33.70 | 84.59 | 74.58 | 54.29 | ✗ | 61.79 |
| S2Seq[GRU] | 60.30 ↑ | 83.40 | 84.53 ↑ | 65.33 | 73.20 | 73.35 |
| GRU | 38.70 | 82.10 | 82.89 | 65.28 | 73.85 | 68.56 |
| S2Seq[Mamba] | 48.80 ↑ | 81.08 | 80.72 ↑ | 61.02 | 71.72 ↑ | 68.67 |
| Mamba | 41.70 | 80.71 | 77.70 | 61.19 | 66.26 | 65.51 |

## D  DST IN GRAPH SETTING

An important aspect of DST-I is its connection to graph theory and therefore S2GNNs. Consider an undirected chain graph where each node is connected to its immediate neighbors. The adjacency

matrix $A$ of this graph is tridiagonal

$$A = \begin{bmatrix} 0 & 1 & 0 & \ldots & 0 & 0 \\ 1 & 0 & 1 & \ldots & 0 & 0 \\ 0 & 1 & 0 & \ldots & 0 & 0 \\ \vdots & \vdots & \vdots & \ddots & \vdots & \vdots \\ 0 & 0 & 0 & \ldots & 0 & 1 \\ 0 & 0 & 0 & \ldots & 1 & 0 \end{bmatrix}$$

and the corresponding degree matrix $D$ can be defined as

$$D = \begin{bmatrix} 1 & 0 & \ldots & 0 \\ 0 & 2 & \ldots & 0 \\ \vdots & \vdots & \ddots & \vdots \\ 0 & 0 & \ldots & 1 \end{bmatrix}$$

The graph Laplacian $L = D - A$ of this chain graph then yields a symmetric, positive semi-definite matrix

$$L = D - A = \begin{bmatrix} 1 & -1 & 0 & \ldots & 0 & 0 \\ -1 & 2 & -1 & \ldots & 0 & 0 \\ 0 & -1 & 2 & \ldots & 0 & 0 \\ \vdots & \vdots & \vdots & \ddots & \vdots & \vdots \\ 0 & 0 & 0 & \ldots & 2 & -1 \\ 0 & 0 & 0 & \ldots & -1 & 1 \end{bmatrix}$$

The S2GNN architecture Geisler et al. (2024) uses the eigenvectors and the eigenvalues of the graph Laplacian to process spectral information. One can calculate these values for the Laplacian of the chain graph above, which would lead to

$$\lambda_k = 2 \left( 1 - \cos \left[ \frac{\pi(k+1)}{N+1} \right] \right); \quad v_k(n) = \sin \left[ \frac{\pi}{N+1}(n+1)(k+1) \right]$$

for its eigenvalues and its eigenvectors. Note that the eigenvectors above follow the exact formula for the DST-I frequencies. This leads to the following conclusion: the application of DST-I in the context of S2Seq is equivalent to the application of S2GNNs over the undirected chain graph of the input.

# E  AUTOREGRESSIVE SPECTRUM

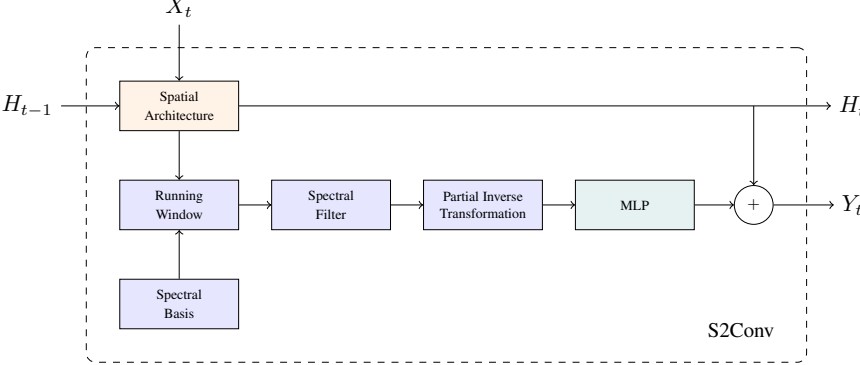

Figure 3: An illustration of the windowed autoregressive variation of the S2Seq architecture. The architecture consists of the spatial branch, the spectral branch, and an MLP. $X_t$ and $Y_t$ denote the input and output at the time $t$. $H_t$ denotes the hidden state of the network at the time $t$.

In this section we define the update rules of the autoregressive S2Seq architecture and take a closer look at the proposed autoregressive spectrum update mechanisms. The updated equation for the

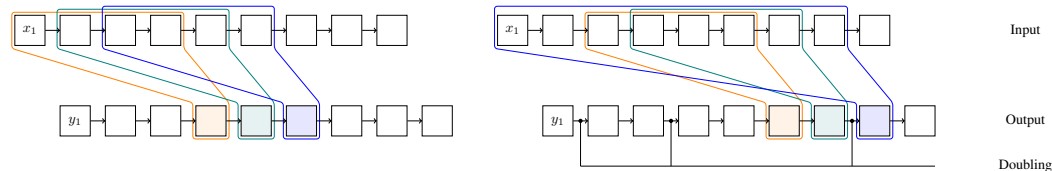

Figure 4: Movement of the spectrally visible suffix given increasing index. The color of the node in the output row corresponds to the same colored region depicting the visible suffix. Left picture corresponds to sliding window. Right picture corresponds to sliding window with doubling.

spatial branch takes the previous hidden state $H^{(t-1)}$ and the current input $X^{(t)}$ and applies $\mathcal{B}_\phi$ from a family of admissible spatial architectures $\mathcal{G}$, i.e.

$$\text{Spatial}^{(t)}(H^{(t-1)}, X^{(t)}) = \mathcal{B}_\phi(H^{(t-1)}, X^{(t)}) \tag{3}$$

Again, let $S^{(t)} = \text{Spatial}^{(t)}(H^{(t-1)}, X^{(t)})$ and let $F$ be the chosen forward transformation. We define $W^{(t)}_{F,N,K}$ as the window function at time $t$. This function performs an efficient running update on the spectral information from the previous $N$ elements using a truncated spectrum of $K$ frequencies calculated using the forward transformation $F$. To make the notation easier to read, we further omit $N$ and $K$ and write $W^{(t)}_F$. To be more precise, $W^{(t)}_F$ is defined as

$$W^{(t)}_F = \mathcal{R}(W^{(t-1)}_F, f_\theta(S^{(t)})) \tag{4}$$

Given a learnable filter $\hat{g}^{(t)}_\vartheta$ and the result of the window function $W^{(t)}_F$, we compute a partial reverse transformation for the current element at time $t$. Its output is then passed through a channel mixing MLP, i.e.

$$\text{Spectral}^{(t)}(S^{(t)}; F, \lambda) = (\text{MLP} \circ F^{-1}[N-1])(\hat{g}^{(t)}_\vartheta(\lambda) \odot W^{(t)}_F) \tag{5}$$

In equation 5, $F^{-1}[N-1]$ denotes the calculation of a partial inverse transformation. Note that the current element at time $t$ is the last element in the window, its index is therefore $N-1$. To be more precise, in case of DFT

$$F^{-1}[N-1](X) = \frac{1}{N} \sum_{m=0}^{K-1} X[m] e^{j2\pi \frac{N-1}{N} m} \tag{6}$$

Similar to the classification task, the output of the spectral branch is multiplied with a learnable parameter $\gamma^{(t)}$

$$H^{(t)} = \text{Spatial}^{(t)}(H^{(t-1)}, X^{(t)}) + \gamma^{(t)} \text{Spectral}^{(t)}(S^{(t)}; F, \lambda) \tag{7}$$

The output $Y^{(t)}$ is then computed based on the result of equation 7. The remaining part of this section dives deeper into the proposed methods for updating the running spectral information.

**Sliding window.** To define the recursive update we turn to the concept of sliding window DFT Springer (1991). In it, the DFT computation is applied to a moving window of a fixed size $N$ that slides along the input signal, requiring $O(LK)$ operations to keep $K$ bins over a signal of length $L$. Its formula, which resembles the one for the Goertzel algorithm, is denoted below (the formula is due to Grado et al. (2017)):

$$X_n[k] = X_{n-1}[k] e^{j2\pi \frac{k}{N}} - x[n-N] + x[n] \tag{8}$$

Figure 4 depicts the movement of the window depending on the index. The obvious trade-off of this approach is the expressivity of the spectrum versus the visibility of earlier elements. If the window size $N$ is picked too small, the spectral branch will not add much information. If it is too large, the spectral information will become less useful in the beginning of a sequence, until enough noise is removed from the window.

**Sliding window with doubling.** To address this problem, we can fully reset the spectrum at exponentially spaced points. Between these points, the spectrum is updated with a regular sliding window. It is easy to see that at all times, the window includes at least half of the previous elements. If we assume a full DFT computation, the overall running time of the algorithm becomes $O(LK + L\log^2 L) = O(L(K + \log^2 L))$. Figure 4 provides an illustration of this approach.

**Exponential window.** Grado et al. (2017) provides an alternative outlook on sliding window DFT and proposes the SWIFT algorithm to maintain an infinite-length Fourier transform with a decay constant $\tau$

$$X_n(\omega) = \sum_{m=-\infty}^{0} e^{m/\tau} x[n+m] e^{-j\omega m} \tag{9}$$

which can also be rewritten as $X_{n+1}(\omega) = e^{-1/\tau} e^{j\omega} X_n(w) + x[n+1]$.

## F RUNNING TIME ANALYSIS

In Table 5 we provide an overview of the running time in big-$O$ notation for the S2Seq[GRU] classification architecture depending on the chosen forward spectral transformation method. In particular, it is worth highlighting that the running time of the spectral branch is sublinear in $L$ in case Sparse FFT is used.

Table 5: Comparison of the running time complexity for S2Seq classification.

| Method | Running Time | Linear in $L$? |
|---|---|---|
| FFT | $O(L\log(L) + LH^2)$ | ✗ |
| Goertzel | $O(LK + LH^2)$ | ✓ |
| Truncated FFT | $O(L\log(K) + LH^2)$ | ✓ |
| Sparse FFT | $O(K\log(L)\log(L/K) + LH^2)$ | ✓ |

It is worth pointing out that real-world performance does not necessarily correspond to the big-$O$ notation guarantees. In particular, the algorithms in the Sparse FFT class usually contain multiple steps and conditional branches, which makes them difficult to implement in a differentiable manner. Furthermore, it is important to note that in our testing the Sparse FFT results were computed assuming a perfect approximation, and the results might differ with a noisy algorithm.

Another potential issue for Sparse FFTs is parallelization and utilization of the available resources. This problem also applies to Truncated FFTs: despite the reduced running time, these algorithms simply skip some elements in the computational butterfly graph, but do not reduce the depth of this graph, which remains $O(\log L)$.

Table 6: Comparison of the running time complexity for S2Seq autoregressive prediction.

| Window Method | FFT Method | Running Time | Linear in $L$? |
|---|---|---|---|
| Sliding Window | N/A | $O(LK + LH^2)$ | ✓ |
| Sliding Doubling | FFT | $O(L\log^2(L) + LK + LH^2)$ | ✗ |
| | Goertzel | $O(LK\log(L) + LK + LH^2)$ | ✗ |
| | Trunc. FFT | $O(L\log(L)\log(K) + LK + LH^2)$ | ✗ |
| | Sparse FFT | $O(K\log^2(L)\log(L/K) + LK + LH^2)$ | ✓ |
| Exp. Window | N/A | $O(LK + LH^2)$ | ✓ |

Table 6 further provides running times of the autoregressive S2Seq[GRU] architecture. It is worth noting that both sliding and exponential windows provide minimal overhead and remain linear in $L$ if

the spatial architecture is linear. The same behavior is demonstrated for the rest of the approaches if the FFT method is sparse (Note: $\log^3(L) = o(L)$, in fact, one can even prove that $\log^K(L) = o(L^a)$ for arbitrary $K > 0$ and $a > 0$). Concerns raised for the classification architecture remain further applicable here.

