# OpenReview forum: "Spatio-Spectral Sequence Processing"
_ICLR.cc/2026/Workshop/GRaM — ICLR 2026 Workshop GRaM Poster_

### Official Review · Reviewer_suvd · 2026-02-14

**Rating:** 6
**Confidence:** 2

**Review:**

**Summary**: The paper proposes adding a learnable spectral branch implemented via a Fourier-like transform (e.g., DFT/FFT, or DST-I for an exact chain-graph Laplacian eigenbasis) to augment a recurrent sequence model. The spectral branch filters features in the frequency domain, maps them back to the time domain, and adds them to the base model’s output via a residual connection.

**Strength**
- The proposed method shows empirical improvements in several tasks of the Long Arena Benchmark.
- The writing is clear and the idea is well presented.

**Weakness**
- I agree that sequences can be modeled as chain graphs, but autoregressive dependencies are directional. That makes an undirected Laplacian basis potentially mismatched; a directed-graph spectral basis would be more complicated. This could be why Text improves only slightly, possibly the gains come mainly from extra capacity (i.e., more parameters) rather than a better inductive bias.
- The method performs forward and inverse time–frequency transforms in every layer. Even if these transforms are fast, truncation or approximation can cause information loss. If the spectral output is added back without careful gating, it could interfere with the base model’s representations.
- I notice that the authors propose sliding-window techniques to handle the autoregressive spectrum. However, the paper does not provide experiments to support these proposals. I therefore question how effective the method would be for text generation tasks.

**Pmlr Suitability:**

NA

---

### Official Review · Reviewer_Qowb · 2026-02-23
**The paper introduces a modular spectral branch for sequence processing, but has inconsistent experimental setups, missing evaluations, and failure to match S4/S5 performance.**

**Rating:** 5
**Confidence:** 3

**Review:**

### Summary
The paper introduces "Spatio-Spectral Sequence" (S2Seq) models, a modular framework designed to augment traditional sequential architectures (e.g., GRUs, Conv1Ds, Mamba) with a parallel spectral branch. This branch leverages spectral transformations (e.g., FFT, Goertzel, or Sparse FFT) and learnable filters to capture global dependencies that local architectures might miss. The authors evaluate these variants on the Long Range Arena (LRA) benchmark and propose a proof-of-concept for extending the method to autoregressive tasks.

### Strengths
* **Well-Grounded Motivation:** The intuition for providing locally-biased models with a "global shortcut" is clear, conceptually sound, and well-supported by recent literature.
* **Modular Flexibility:** The proposed framework is versatile, demonstrating how a spectral branch can theoretically be bolted onto a variety of existing base architectures.
* **Exploration of Sublinear Methods:** The inclusion and theoretical analysis of Sparse FFT and Goertzel algorithms show a thoughtful attempt to address the computational bottlenecks of standard global attention.

### Weaknesses
* **Overstated Claims and Performance Gaps:** The claim that truncated spectra can match full FFT calculations is overstated. S2Seq variants fail to bridge the gap to state-of-the-art State Space Models (SSMs) like S4/S5, with a 2% accuracy gap remaining on ListOps and massive deficits on Image/Path tasks.
* **Inconsistent Improvements:** Table 3 shows the baseline Mamba outperforming S2Seq[Mamba] on the Image task (61.19 vs 61.02). This directly contradicts the narrative of "consistent improvement" and raises unaddressed questions about potential interference between the spatial and spectral representations.
* **Lack of Practical Efficiency Metrics:** While Table 5 highlights theoretical $O(\cdot)$ sublinear time for Sparse FFT, this assumes a "perfect approximation." Without a practical **Wall-clock time vs. Accuracy** plot, it is impossible to verify if this model is actually faster or more memory-efficient than S4/S5 in real-world scenarios.
* **Architectural Choices:** In Equation 1, the spectral branch takes the *output* of the spatial branch as its input. This creates a sequential dependency where the global branch is merely filtering locally-processed features, potentially limiting its true global receptive capabilities compared to a strictly parallel structure acting on raw input $H^{(l-1)}$.
* **Methodological and Reporting Flaws:**
    * The results appear to rely on single runs without standard deviations, which is problematic given LRA's known sensitivity to initialization.
    * The "Avg." column in Table 1 artificially inflates performance by excluding the failed Conv1D-Path result. Failed runs are significant data points and must be factored into the mean.
    * Mamba, a primary modern baseline, is inconsistently relegated to Appendix B instead of Table 1.
* **Technical Ambiguities:**
    * Notation is incomplete (e.g., $g$ and $f$ are undefined in Equation 1).
    * The performance dip in Figure 2(b) for magnitude-based frequency selection at $K=8$ and $K=16$ should be better addressed.
    * The claim in Appendix E that larger windows cause information to "dilute" is unclear and requires clarification.

### Overall Recommendation
The paper presents an interesting modular extension, but the empirical results do not yet justify the added complexity when compared to established, fully integrated SSMs. The reliance on theoretical efficiency over practical wall-clock benchmarks, combined with overstated performance claims and methodological reporting issues, weakens the submission.

**Pmlr Suitability:**

NA

---

### Meta-Review · Area_Chair_FM3i · 2026-02-25

**Decision:**

Accept

**Metareview:**

The idea of the paper is novel and relevant to GRaM. The experiments show signs of life. The reviewers point out some legitimate shortcomings in the experiments, which the authors should address. However, since the intention of a tiny paper is to capture interesting, in-progress work that need not be SOTA yet, I recommend acceptance.

**Relevance To Proceedings:**

Tiny paper — does not apply

**Relevance To Workshop:**

Yes — suitable for GRaM

---

### Decision · Program_Chairs · 2026-03-02

Accept (Poster)